# Synthesis of Oxidized 3β,3′β-Disteryl Ethers and Search after High-Temperature Treatment of Sterol-Rich Samples

**DOI:** 10.3390/ijms221910421

**Published:** 2021-09-27

**Authors:** Adam Zmysłowski, Jerzy Sitkowski, Katarzyna Bus, Katarzyna Michalska, Arkadiusz Szterk

**Affiliations:** 1Spectrometric Methods Laboratory, National Medicines Institute, Chełmska 30/34, 00-725 Warsaw, Poland; k.bus@nil.gov.pl; 2Falsified Medicines and Medical Devices Department, National Medicines Institute, Chełmska 30/34, 00-725 Warsaw, Poland; j.sitkowski@nil.gov.pl; 3Department of Synthetic Drugs, National Medicines Institute, Chełmska 30/34, 00-725 Warsaw, Poland; k.michalska@nil.gov.pl; 4Center for Translationale Medicine, University for Life Sciences, Nowoursynowska 100, 02-797 Warsaw, Poland; szterkarkadiusz@gmail.com; 5Transfer of Science Sp. z o.o., Janiszowska 14, 02-264 Warsaw, Poland

**Keywords:** cholesterol, β-sitosterol, stigmasterol, oxidized 3β,3′β-disteryl ethers, liquid chromatography, mass spectrometry

## Abstract

It was proven that sterols subjected to high-temperature treatment can be concatenated, which results in polymeric structures, e.g., 3β,3′β-disteryl ethers. However, it was also proven that due to increased temperature in oxygen-containing conditions, sterols can undergo various oxidation reactions. This study aimed to prove the existence and perform quantitative analysis of oxidized 3β,3′β-disteryl ethers, which could form during high-temperature treatment of sterol-rich samples. Samples were heated at 180, 200 and 220 °C for 0.5 to 4 h. Quantitative analyses of the oxidized 3β,3′β-disteryl ethers were performed with liquid extraction, solid-phase extraction and liquid chromatography coupled with mass spectrometry. Additionally, to perform this analysis, the appropriate standards of all oxidized 3β,3′β-disteryl ethers were prepared. Eighteen various oxidized 3β,3′β-disteryl ethers (derivatives of 3β,3′β-dicholesteryl ether, 3β,3′β-disitosteryl ether and 3β,3′β-distigmasteryl ether) were prepared. Additionally, the influence of metal compounds on the mechanism of ether formation at high temperatures was investigated.

## 1. Introduction

It was already proven that oxysterols and oxyphytosterols are formed through the oxidation of sterol molecules and possess various functions or influences in the human body. The aspect of oxidized sterol molecules and their connection with diseases, such as atherosclerosis, was already studied and discussed [1,2].

The various reactions of sterol molecules during high-temperature treatment not only produce oxysterol but also sterol polymers. Struijs et al. demonstrated the formation of stigmasterol polymers after heating at 180 °C [3]. Based on the molecular mass obtained by high-resolution mass spectrometry (HRMS), one of the found dimers was suggested to consist of two stigmasterol moieties linked with ether bonds, which could correspond to the structure of 3β,3′β-distigmasteryl ether. Additionally, using gas chromatography coupled with mass spectrometry (GC-MS), we recently proved the formation of 3β,3′β-disteryl ethers in sterol-rich samples after high-temperature treatment [4]. However, in addition to the polymerization reaction, formed ethers could be additionally oxidized or, as also presumed, formed oxysterols during heat treatment could react with native sterol to form oxidized 3β,3′β-disteryl ethers because only one sterol particle needs the 3β-hydroxyl and double bond at positions 5–6 [4]. This possibility was confirmed by Struijs, who found substances based on molecules that were 3β,3′β-distigmasteryl ether oxidized at position 7 to a ketone or hydroxyl moiety. Similar studies were performed by Sosińska et al. using heated samples of β-sitosterol [5]. Available data indicated that 3β,3′β-disteryl ether was the most abundant compound in the nonpolar fraction, while 7-keto 3β,3′β-disteryl ether was the major dimer in the mid-polar fraction. A 7-hydroxy 3β,3′β-disteryl ether was identified as a prominent dimeric component in the complex polar phytosterol degradation products. 7-Ketosteryl-sterol ethers were also recently synthesized; however, there was no confirmation of their presence after sterol heat treatment [4].

The aim of the present study was to evaluate the possibility of forming oxidized sterol dimers—3β,3′β-disteryl ethers—during the thermal treatment of sterols. For this purpose, synthesis methods were developed, and an attempt was made to prepare different oxidized 3β,3′β-disteryl ethers. After proper identification and characterization by NMR and HRMS, synthesized compounds were used as standards for development and validation of the analytical method for quantifying the compounds after thermal treatment of sterol-rich samples. Additionally, the influence of metal compounds on the mechanism of ether formation at high temperatures was investigated.

## 2. Results

### 2.1. Synthesis of the Sterol Derivatives

The synthesis route is presented in Figure 1. The formation of 3β,3′β-disteryl ethers was achieved first with the reaction of sterols with the MK10 catalyst in CH_2_Cl_2_. After obtaining 3β,3′β-disteryl ethers (1, 2 and 3), they were subjected to direct oxidation by the chromium oxide/3,5-dimethylpyrazole complex in CH_2_Cl_2_ to yield 7-ketosteryl-sterol ethers (4, 5 and 6). However, our previous study of the reaction mechanism of ether formation proved that 7-ketosteryl-sterol ethers can also be obtained using sterol with 7-ketosterol in reaction with MK10 in CH_2_Cl_2_ [4]. To achieve 7-ketosterols, first, the 3β-hydroxyl groups of native sterols were protected as acetate (22, 23 and 24), and the sterol moiety underwent an oxidation reaction by the chromium oxide/3,5-dimethylpyrazole complex in CH_2_Cl_2_ to give 7-keto derivatives 25, 26 and 27. At this stage, a deprotection reaction in alkaline medium readily led to 7-ketosterols (28, 29 and 30). The reaction of the obtained 7-ketosterols and native sterols in CH_2_Cl_2_ with the MK10 catalyst resulted in 7-ketosteryl-sterol ether (4, 5 and 6). Double 7-keto 3β,3′β-disteryl ethers (7, 8 and 9) were synthesized by increasing the molecular ratio of the chromium oxide/3,5-dimethylpyrazole complex in the oxidation of 3β,3′β-disteryl ethers. Double oxidized 3β,3′β-disteryl ethers could not be obtained through the reaction of 7-ketosterol with MK10. Most likely, the addition of the 7-keton moiety inhibited the formation of an i-steroid intermediate ion, which is required to form an ether bond. Prepared 7-ketosteryl-sterol ethers (4, 5 and 6) were then subjected to reduction by sodium borohydride or L-selectride to yield 7-hydroxysteryl-sterol ether in the α- (10, 11 and 12) or β- (13, 14 and 15) form. The epoxy derivatives were obtained using mCPBA or synthetized Cu(MnO_4_)_2_ as was used earlier [6], which gave 5,6α-epoxysteryl-sterol ethers (16, 17 and 18) and 5,6β-epoxysteryl-sterol ethers (19, 20 and 21).

Additionally, cholesta-3,5-diene (**31**) was obtained by reaction with ZnSO_4_ at 220 °C. The product was isolated as a mixture of isomers: cholesta-3,5-diene:cholesta-2,4-diene at a ratio of 5:1, which we could not separate; therefore, it was used as-is.

### 2.2. Metal Content in MK10 Catalyst and in Chosen Biological Samples

To verify the metal content in the catalyst used for the reaction (MK10) and chosen biological samples, the ICP-MS and AAS methods were used. The MK10 sample was mineralized in a mixture of acids, hydrochloric acid, nitric acid, and hydrofluoric acid, and analysed. The biological samples were first burned to ashes and diluted with nitric acid. Using ICP-MS for each metal ion, calibration curves were prepared from 0.5 to 50 ppb. Fifty-four determined calibration curves are presented in Appendix A. The samples were appropriately diluted to fit the range of the calibration curves. Because of the high concentration of sodium, calcium and potassium in MK10, the contents of the metals were measured using the AAS method instead of ICP-MS. The results obtained for the MK10 catalyst and chosen samples are presented in Appendix A. As expected, the highest contents in MK10 were aluminium, iron and magnesium [4] and were 4.13, 1.23 and 0.55%, respectively.

The metals in highest concentration in butter and cod-liver oil were sodium and potassium, 1.1 and 3.5 ppm for butter and 0.5 and 0.4 ppm for cod-liver oil, respectively. Based on the results, the plant oils had a relatively higher concentration of the most common metals. The rapeseed oil was measured with high concentrations of magnesium (7.1 ppm), calcium (4.0 ppm), potassium (0.8 ppm), sodium (0.7 ppm), iron (0.4 ppm) and manganese (0.3 ppm). Corn oil had high concentrations of sodium (2.1 ppm), magnesium (1.5 ppm), potassium (1.2 ppm), boron (1.2 ppm), iron (0.3 ppm) and calcium (0.3 ppm). Additionally, in all samples, zinc was measured at concentrations of 0.18 ppm for butter, 0.04 ppm for cod-liver oil, 0.31 ppm for rapeseed oil and 0.12 ppm for corn oil. Additionally, corn oil also had at least double the concentration of copper (0.10 ppm), in contrast to the other samples.

### 2.3. Validation of LC-CAD and LC-MS Methods

To study the influence of metal catalysts on the formation of 3β,3′β-disteryl ethers, the LC-CAD method was developed. In addition to the 3β,3′β-dicholesteryl ether, cholesta-3,5-diene and cholesterol were determined in each sample. The CAD chromatogram of the separation is presented in Figure 2. The limits of detection (LODs) and limits of quantification (LOQs) for the LC-CAD method were calculated using the standard deviation of the response and the slope of the calibration curve, which were near to the lower concentrations, where the calibration curves were linear. The results are presented in Appendix A. To analyse the oxidized 3β,3′β-disteryl ethers, LC-MS with APCI ionization was used. The chromatogram of separated oxidized 3β,3′β-dicholesteryl ethers is presented in Figure 3. The linearity was evaluated, and the LODs and LOQs were calculated based on the standard deviation of the response and the slope of the analytical curve. Recovery tests for each compound were conducted using three concentration levels: low concentration (50 ng/mL), medium concentration (100 ng/mL) and high concentration (250 ng/mL). The LOD, LOQ, and recovery at different levels and linearity range of the developed method for oxidized 3β,3′β-dicholesteryl ethers are shown in Appendix A, and those for 3β,3′β-disitosteryl and 3β,3′β-distigmasteryl ethers are shown in Appendix A.

### 2.4. Analysis of High-Temperature Treatment Samples

Using the prepared and validated LC-CAD method, 3β,3′β-dicholesteryl ether, cholesta-3,5-dien and cholesterol were analysed in samples that were subjected to treatment at various temperatures with different inorganic compounds. The content of each compound is expressed as per gram of cholesterol in Table 1, whereas cholesterol is presented as a percent of the initial concentration. As expected in each sample, the cholesterol concentration decreased with heating time. For aluminium chloride, the cholesterol concentration at 180 °C decreased rapidly, and at 200 and 220 °C, even after 0.5 h of heating, the cholesterol could not be quantified. Similar to the result for 220 °C, the measured cholesterol decreases and cannot be quantified in magnesium after 1 h and for copper samples after 0.5 h of heating time. The highest concentration of 3β,3′β-dicholesteryl ether was acquired for copper salt at 180 °C, with a maximum value after 3 h of heating of 103.15 ± 29.20 mg/g. In addition to the aluminium samples, the concentration of measured 3β,3′β-dicholesteryl ether increased with heating time at 180 and 200 °C. At 220 °C, in addition to zinc, samples were found with decreasing ether concentration with heating time. The calcium samples after heat treatment were not found with 3β,3′β-dicholesteryl ether at any of the chosen temperatures. The highest concentration of cholesta-3,5-diene was measured in the zinc samples, with a maximum value after a 4 h heating time of 220 °C—320.79 ± 6.19 mg/g. At 180 °C, the concentration of cholesta-3,5-diene increased with heating time in the copper, magnesium, zinc and aluminium samples. The same increasing concentration with heating time was also observed in copper, magnesium and zinc samples at 200 °C and calcium and zinc samples at 220 °C. However, a decrease in the concentration over time was observed for aluminium samples at 200 °C and copper, magnesium and aluminium samples at 220 °C.

Using a validated LC-MS method for analysing oxidized 3β,3′β-dicholesteryl ether, cholesterol samples with different inorganic salts were analysed. The contents of each compound are presented in Table 1. The formation of 7-ketocholesteryl-cholesterol (7-kCh-Ch) ether was measured with the highest concentration in magnesium samples after a 3 h heating time of 200 °C—5.65 ± 1.78 µg/g. An increasing concentration with heating time was measured with copper and zinc samples at 180 °C and in magnesium samples at 200 °C. The remaining samples, especially at a heating temperature of 220 °C, were observed with decreasing concentration with time of heating. Additional oxidized 3β,3′β-dicholesteryl ether was found in magnesium, aluminium and copper samples with 7-hydroxy and 5,6-epoxy derivatives. The highest concentration of 7α-hydroxycholesteryl-cholesterol (7α-hCh-Ch) ether (1.18 ± 0.47 µg/g) was measured in the magnesium sample after heating for 4 h at 200 °C. Even double oxidized ketone 3β,3′β-dicholesteryl ether was found in the magnesium sample at 200 and 220 °C, with a highest concentration of 0.17 ± 0.05 µg/g.

Using a validated method to analyse oxidized 3β,3′β-disteryl ethers, the biological origin samples were analysed after heating at different temperatures. The results are presented in Table 2. The concentration is presented as ng per gram of the chosen sample. 7-kCh-Ch was measured in butter and cod-liver oil only at 180 and 200 °C. The highest concentration of this ether was measured in butter after 4 h of heating at 180 °C (40.53 ± 13.73 ng/g). However, for the plant origin samples, the only oxidized 3β,3′β-disteryl ether quantified was 7-ketositosteryl-sitosterol ether (7kSito-Sito) in rapeseed oil. The highest concentration measured was 2399.33 ± 560.95 ng/g, which was found in the sample after 0.5 h of heating at 180 °C.

## 3. Discussion

### 3.1. Chromatography Methods

The very low polarity of the 3β,3′β-disteryl ethers and oxidized 3β,3′β-disteryl ethers requires a different approach to analysis by chromatography and mass spectrometry. Therefore, based on our previously reported GC-MS method, which showed that analysis of 3β,3′β-disteryl ethers is possible with gas chromatography, an attempt was made to use this technique [4]. However, the addition of oxygen to the sterol structure increased the boiling point of the synthesized compounds, which already in the case of 3β,3′β-disteryl ether resulted in the use of appropriate high-temperature chromatography. Moreover, in addition to the hydroxyl moiety, the ketone moiety and epoxy oxygen are not suitable for derivatization, e.g., silanization, which is a common approach in GC analysis. Therefore, for the analysis of oxidized 3β,3′β-disteryl ethers, liquid chromatography coupled with mass spectrometry was needed. Examples of separated oxidized 3β,3′β-dicholesteryl ethers are presented in Figure 3. The typical ion source used for sterol analysis coupled with liquid chromatography is APCI. The only compounds that were stable for the pseudomolecular ions present were the ketone derivatives. Additionally, the compounds gave many low abundances of fragmentation ions with collision gas; therefore, reaction monitoring could not be used with high sensitivity. For that capability, we decided that the analysis should be concluded with SIM analysis. The chromatography of these nonpolar compounds requires using nonpolar solvents. Unfortunately, any addition of such a solvent in the gradient programme lowered the sensitivity. After testing, it was concluded that MeOH and CH_2_Cl_2_ would be chosen for the composition of the mobile phase. For instance, the use of ACN instead of MeOH resulted in narrower peaks; however, in this case, ACN, because of its substantial decrease in ionization, cannot be used in LC-MS with an APCI source. Therefore, the ACN and CH_2_Cl_2_ solvents were used in the LC-CAD method, which gave higher signals than using MeOH. Additionally, it is worth mentioning that the CAD response is commonly described as approximately linear or quasi-linear over a range of 1.5 to 2 orders of magnitude but non-linear over a wider range. As in this case, the chosen range of prepared curves resulted in a non-linear response; therefore, the equation used to quantify the concentration of compounds was a second-order polynomial.

### 3.2. Formation of 3β,3′β-Disteryl Ethers

Based on previous research, the conditions that need to be met to form 3β,3′β-disteryl ethers are 3β-hydroxyl and 5–6 double bonds in one of the sterol molecules [4]. Heating sterol standards themselves can result in 3β,3′β-disteryl ether formation; however, it was already proven that this reaction can occur in higher yields when copper ions are present. It was reported that 3β,3′β-disteryl ethers were prepared using a reaction at high temperature with copper sulfate as a catalyst [3,5,7,8], which suggests that the catalyst, such as in this case metal ions, could influence the formation of an ether bond. Additionally, based on the results of the metal content of the MK10 catalyst, we wanted to verify whether any other metal ion present while heating sterol can induce the formation of 3β,3′β-disteryl ether. The metal ions tested were calcium, copper, magnesium, zinc and aluminium used as sulfate or chloride salts. Copper was chosen because it has already been proven that this transition metal has various impacts on the oxidation and 3β,3′β-disteryl formation of sterols. Additionally, copper is also used as a catalyst for the study of atherosclerosis through the oxidation of sterol-containing low-density lipoprotein (LDL) [9,10]. Zinc was chosen as an additional transition metal, which was measured by ICP-MS at relatively high concentrations in the chosen biological samples. Interestingly, it was recently assumed that zinc has a protective oxidation role in the modification of cholesterol in patients with type 1 diabetes [11]. However, it can play the opposite role when present on food compounds during high-temperature treatment. Magnesium and calcium were chosen based on general occurrence. However, aluminium was chosen based on the results acquired from the MK10 catalyst, which, based on previous research, is probably mainly responsible for ether formation during the synthesis, which was also confirmed by ICP-MS as the most concentrated metal found in this catalyst.

To analyse the metal influence on the formation of 3β,3′β-disteryl ethers and their oxidized forms, samples with cholesterol after temperature treatment were analysed using two chromatographic methods. In addition to the concentration measured using the LC-MS method for oxidized 3β,3′β-disteryl ethers, the LC-CAD method was developed for determination of cholesta-3,5-diene, 3β,3′β-dicholesteryl ether and cholesterol in each sample after heating at the chosen temperatures. Cholesta-3,5-diene is one of the main cholesterol transformation molecules because of its dehydration. At 180 and 200 °C, the 3β,3′β-dicholesteryl ether and cholesta-3,5-diene concentrations increased with heating time for copper, zinc, magnesium and aluminium. The highest concentration of 3β,3′β-dicholesteryl ether was quantified with copper after 4 h of heating at 180 °C. Increasing concentrations with heating time were also present at 220 °C with zinc with 3β,3′β-dicholesteryl ether and cholesta-3,5-diene. However, with copper, magnesium and aluminium, the heating decreased the concentration of both compounds with time. These results were probably obtained because of extensive reactions of the formed molecules. An example of this reaction is the fragmentation of the sterol molecule, which results in lower molecular weight compounds [12]. A decrease in concentration was also obtained in our previous research by quantifying the 3β,3′β-disteryl ethers in high-sterol-containing samples [4]. Theoretically, formed 3β,3′β-dicholesteryl ether at these high temperatures can undergo the dehydration reaction to form two cholesta-3,5-diene molecules, which in addition to the dehydration reaction of cholesterol could explain the increased concentration of cholesta-3,5-diene and decrease in 3β,3′β-dicholesteryl ether. Using calcium as a catalyst resulted in the formation of only cholesta-3,5-diene at 200 and 220 °C, with no 3β,3′β-dicholesteryl ether detected; however, surprisingly, cholesta-3,5-diene was found with 7-kCh-Ch ether. The maximum concentration of measured 7-kCh-Ch was obtained in the magnesium sample after 3 h of heating at 200 °C. Unfortunately, no 3β,3′β-dicholesteryl ethers or their oxidized forms were found in the cholesterol-heated samples (without any catalyst addition) at all temperatures (data not shown), which did not comply with previously published data [13]. However, these results indeed validate the assumption that during heating, a catalyst needs to be present, for example, as metal ions, to form the studied compounds. The most interesting results considering oxidized 3β,3′β-dicholesteryl ethers were obtained using magnesium and aluminium compounds. As expected, the oxidized ether with the highest concentration was 7-kCh-Ch. In aluminium and magnesium samples at 180 and 200 °C, additional synthesized oxidized 3β,3′β-dicholesteryl ethers were detected and quantified (hydroxyl and epoxy derivatives). Even in the magnesium sample after 4 h of heating at 200 °C, all synthetized 3β,3′β-dicholesteryl ethers were quantified. Similar to 3β,3′β-dicholesteryl ether, the concentration of oxidized 3β,3′β-dicholesteryl ethers increased with the temperature and incubation time [4]. However, at 220 °C, the results obtained at 0.5 h and 1 h suggest that oxidation or ether formation occurs; however, continuously over time, the compounds degrade.

As we assumed, these oxidized ether compounds are formed from cholesterol and 7-ketocholesterol or due to oxidation of the formed 3β,3′β-dicholesteryl ether. The same is possible with cholesterol and 5,6-epoxycholesterol and theoretically possible with a 7-hydroxy derivative; however, the 7-hydroxy moiety could also react to form the ether with oxygen at position 7. Based on acquired data on heating cholesterol with metal compounds, it was concluded that oxidized 3β,3′β-disteryl ethers can indeed be formed during heating. However, the concentration of the oxidized 3β,3′β-disteryl ethers was low, and it was assumed that in the chosen biological origin samples, the concentration would be similarly low or below the detection limit.

Only one oxidized 3β,3′β-dicholesteryl ether, 7-kCh-Ch, was found in butter and cod-liver oil. The concentration of measured 7-kCh-Ch in butter at 180 °C increased from 3 to 4 h. The obtained concentrations correspond to obtained previous results for formed 3β,3′β-dicholesteryl ether [4]. With the increased concentration of 3β,3′β-dicholesteryl ether over time, the 7-kCh-Ch concentration also increased, which suggested that the 3β,3′β-dicholesteryl ether was slowly oxidized with heating. The same results were obtained at 200 °C for butter, and the highest concentration corresponded to the highest concentration of 7-kCh-Ch at 4 h of heating [4]. Interestingly, the highest concentration of 3β,3′β-dicholesteryl ether measured in butter at 220 °C does not correspond to finding any 7-kCh-Ch in the samples. Based on the results obtained from cholesterol heated with metal compounds, where the 3β,3′β-dicholesteryl ether and 7-kCh-Ch concentrations decreased rapidly with heating time, it can be concluded that sterols or formed 3β,3′β-dicholesteryl ethers are more prone to other reactions. Surprisingly, 7-kCh-Ch was also quantified in 0.5 h at 200 °C in cod-liver oil, which could also suggest that the formation of these compounds can be more complex and that there are other additional factors that should be considered; however, at this point, these factors are elusive. This also suggests the results obtained for plant origin samples, where 7-kSito-Sito in rapeseed oil was the only 3β,3′β-disteryl ether that was found. However, the highest concentration was observed at 0.5 h at 180 °C, decreased rapidly with heating time and was almost 100-fold higher than the other concentrations measured.

Based on acquired results of biological samples from ICP-MS, it could be possible that the sum of all metal contents could catalyse the formation of 3β,3′β-disteryl ethers or their oxidized form. In each sample, as expected, the metal content differed slightly. The overall content of metals in rapeseed oil could contribute to the measured concentration of oxidized 3β,3′β-disitosteryl ether compared to corn oil, where synthesized compounds were not found. For instance, the measured concentrations of magnesium and zinc, which were found to catalyse ether formation and/or its oxidation, were almost 5- and 2-fold higher in rapeseed oil than in corn oil, respectively. It should be mentioned that samples were heated in open glass vials; however, changing any metal container (which mostly occurs during cooking) could have an impact on the formation of ether because of surface catalysis. Additionally, the chosen samples are mixtures of various substances, which are far from ideal samples, such as single-heated sterols, and their influence is not always simple to predict. For instance, the presence of different fatty acids, especially unsaturated fatty acids, probably influences oxidation and ether formation. It seems that the higher concentration of unsaturated fatty acids in vegetable oils causes oxidation to be the primary reaction occurring through heat treatment. However, it is also possible that the ether formation reaction occurs independently of unsaturated fatty acid oxidation. Nevertheless, the obtained results confirmed that oxidized 3β,3′β-disteryl ethers can indeed form in high-temperature-treated sterol-rich samples.

## 4. Materials and Methods

### 4.1. Materials

All solvents for synthesis were of HPLC-grade quality. Isopropanol (IPA), tetrahydrofuran (THF), dichloromethane (CH_2_Cl_2_), chloroform (CHCl_3_), n-hexane (n-Hex), tert-butanol (t-BuOH), methanol (MeOH), diethyl ether (Et_2_O), ethyl acetate (EtOAc), methyl tert-butyl ether (MTBE), pyridine and acetone were purchased from Merck Millipore. CuSO_4_, CaSO_4_, ZnSO_4_·H_2_O, MgCl_2_ and AlCl_3_·6H_2_O were purchased from Sigma Aldrich. Synthesis reagents were purchased from Sigma-Aldrich Company, except stigmasterol (Cayman Chemicals, Ann Arbor, MI, USA). Deuterated chloroform-d_1_ (99.95 atom % D) was obtained from Dr Glaser AG Basel. Suprapur grade 65% nitric acid (HNO_3_) was purchased from Merck Millipore. Multielement solutions at a concentration of 10 mg/L each of Ag, Al, As, Au, B, Ba, Be, Ca, Cd, Ce, Co, Cr, Cs, Cu, Dy, Er, Eu, Fe, Ga, Gd, Hg, Ho, Ir, K, La, Li, Lu, Mg, Mn, Mo, Na, Nd, Ni, Os, Pb, Pd, Pr, Pt, Rb, Rh, Ru, Sb, Se, Sm, Sn, Sr, Th, Tl, Tm, V, U, V, Yb, and Zn were purchased from INORGANIC Ventures (Christiansburg, VA, USA). The purities of plasma gas (argon) and cell gas (helium) were greater than 99.999%. Rapeseed, corn oil (cold-pressed) and butter (based on the label: 82% fat content) were purchased from a local market. Cod-liver oil was purchased from a local pharmacy. Strata silica SI-1 cartridges (500 mg, 3 mL, 55 µm particle size, 70 Å pore size) were acquired from Phenomenex (Torrance, CA, USA). Thin-layer chromatography (TLC) was performed on precoated silica gel plates (Merck 60 PF254). Visualization of compounds on TLC plates was achieved by ultraviolet (UV) (254 nm) light detection and/or iodine staining. Dry column vacuum chromatography (DCVC) was performed using Merck silica gel 60 [14]. Montmorillonite K 10 (MK10) was dried for 2 h at 125 °C before the reaction. Pure β-sitosterol was obtained according to a previously published method [6]. Each compound obtained in pure crystal form was stored at 4 °C.

### 4.2. Chemical Synthesis

The synthesis methods of all used compounds (**1–31**), along with the ^1^H and ^13^C NMR data, can be found in the Appendix A. The synthesis route is presented in Figure 1.

### 4.3. Standard and Sample Preparation

#### 4.3.1. Sample Preparation of Cholesterol with Inorganic Salts

A total of 250 mg of cholesterol was mixed and ground together into a fine powder with a mortar and pestle with 50 mg of each inorganic compound chosen for the study (as Table 1 indicates). After appropriate mixing, 30 mg of the sample was weighed into 2 mL vials. The top was capped with a rubber stopper, and the samples were placed in an oven for the time and temperature indicated in Table 1. The samples after cooling were stored at 4 °C. Before the analysis, samples were dissolved in 1 mL of THF and filtered through a 0.22 µm PTFE filter. For analysis with the LC-CAD method, samples were used as is, and for the analysis with LC-MS, the samples were diluted with MeOH:THF solution (4:1 *v*/*v*) accordingly to fit the calibration curves with internal standard addition (7-ketostigmasteryl-stigmasterol ether) to a final concentration of 250 ng/mL.

#### 4.3.2. Sample Preparation of Biological Origin Samples

Samples were chosen based on their high native sterol concentration [15,16]. Thus, butter and cod-liver oil were chosen for analysing oxidized 3β,3β’-dicholesteryl ethers. Crude rapeseed and corn oil were chosen for analysing oxidized 3β,3β’-disitosteryl and 3β,3β’-distigmasteryl ethers. The butter before heating was clarified by melting at a low temperature (60 °C) and removing undissolved sediment.

Prepared samples were oven heated in open glass vials for the time and temperature indicated in Table 2. The samples were allowed to cool and then subject to saponification. A portion of each sample (1.0 g) after heat treatment was mixed with 15 mL of 1 M KOH in MeOH and left for 24 h in the dark. Next, all saponified samples were extracted at least twice with 20 mL of n-Hex each time, and the solvent was evaporated using a rotavapor. Samples were redissolved in 5 mL of n-Hex and extracted at least 3 times with 2 mL of MeOH. After the extraction solvent was evaporated in a stream of nitrogen and redissolved in 1 mL of n-Hex, samples were subjected to purification using solid-phase extraction.

Strata silica SI-1 cartridges were conditioned with 5 mL of n-Hex. The 1 mL of n-Hex phase was applied on SPE. The cartridge was rinsed with 5 mL of n-Hex:CH_2_Cl_2_ (10:1 *v*/*v*). Oxidized 3β,3β’-disteryl ethers were eluted with 10 mL of n-Hex:EtOAc 6:1 (*v*/*v*) solution. The solvent was evaporated under a stream of N_2_ at 30 °C. The residue was dissolved in 1 mL of MeOH:THF solution (4:1 *v*/*v*).

#### 4.3.3. Sample Preparation for NMR Analysis

Samples from 10 to 20 mg were dissolved in 600 µL of deuterated chloroform-d_1_.

#### 4.3.4. Sample Preparation for Inductively Coupled Plasma Mass Spectrometry (ICP-MS) and Atomic Absorption Spectrometry (AAS) Analysis

Approximately 1 g of each sample (butter or oils) was inserted directly into quartz crucibles and burned to ash. After the burning procedure, the samples were diluted to a final volume of 100 mL with 1% HNO_3_.

Approximately 1 g of MK10 sample was inserted directly into a PTFE-closed vessel, and 1 mL of concentrated HCl, 2 mL of 48% HF and 3 mL of 65% HNO_3_ were added. The mineralization of the samples was performed for 24 h under atmospheric pressure. After the procedure, the samples were diluted to a final volume of 100 mL with water. For analysis, the samples were diluted to fit the calibration curve concentrations.

### 4.4. Chromatographic Methods

#### 4.4.1. LC-CAD Method

A UHPLC Ultimate 3000 (Dionex Thermo Fisher Scientific, Sunnyvale, CA, USA) system consisting of a pump, degasser, autosampler, and column heater instrument was used. Data processing was performed with Chromeleon 7.0 and Chromeleon Validation ICH software (Dionex). The detector used was a charged aerosol detector (CAD)—Corona Veo Thermo Scientific. Chromatographic separation was conducted using a Waters Acquity CSH C18 150 × 2.1 mm, 1.7 µm column. The work was conducted in a gradient system (phase A using ACN and phase B: ACN:CH_2_Cl_2_ 40:60 *v*/*v*). The following gradient system was applied: 0–1 min 20% B, 1–17 min 100% B, 17–20 min 100% B, 20–25 min 20% B and 25–34 min 20% B, balancing columns until the initial conditions were restored. Chromatographic separation was conducted with a constant flow of the mobile phase (400 μL/min) at 30 °C. The injection volume on the column was 1 μL.

#### 4.4.2. LC-MS Method for Oxidized 3β,3′β-Disteryl Ethers

A UHPLC 1260 infinity (Agilent, Santa Clara, CA, USA) system consisting of a pump, degasser, autosampler and column heater was used. To determine the peak elution order, an LC-MS 6460 triple-quadrupole mass spectrometer from Agilent was employed. Atmospheric pressure chemical ionization (APCI) in positive ionization in selected ion monitoring (SIM) was used. The optimised parameters of the mass spectrometer for each compound are presented in Appendix A. Chromatographic separation was conducted with the use of a Waters Acquity BEH C18 100 × 2.1 mm, 1.7 µm column, in a gradient system (phase A using MeOH and phase B: MeOH:CH_2_Cl_2_ 40:60 *v*/*v*). The following gradient system was applied: 0–1 min 20% B, 1–11 min 100% B, 11–13 min 100% B, 13–15 min 20% B and 15–21 min 20% B, balancing columns until the initial conditions were restored. Chromatographic separation was conducted with a constant flow of the mobile phase (500 µL/min) at 30 °C. The injection volume on the column was 5 μL.

#### 4.4.3. LC-QTOF-MS Methods

The spectrometry method conditions were the same as those previously published [4]. A UHPLC Ultimate 3000 (Dionex Thermo Fisher Scientific, Sunnyvale, CA, USA) system consisting of a pump, degasser, autosampler, and column heater instrument was used. Data processing was performed with Chromeleon 6.8 and Chromeleon Validation ICH software (Dionex, Sunnyvale, CA, USA). To determine the peak elution order, a maXis 4G mass spectrometer from Bruker Daltonic (Billerica, MA, USA) was used. The QTOF settings were atmospheric pressure chemical ionization (APCI) in positive ion mode, nebulizer at 2.0 bar, dry gas (nitrogen) flow rate of 4.0 L/min, dry heater at 200 °C, vaporizer temperature of 450 °C, capillary voltage of 3000 V, corona discharge of 3000 nA and endplate offset of −500 V. MS data were recorded in full scan mode (from 100 to 1600 *m*/*z*). The mass spectrometer was used in high-resolution mode (R = 60,000), and an internal calibrant (APCI/APPI calibrant) was used to make a precise mass measurement. Chromatographic separation was conducted using a Waters Acquity BEH C18 100 × 2.1 mm, 1.7 µm column in a gradient system (phase A: MeOH, Phase B: MeOH:CH_2_Cl_2_ 2:3 *v*/*v*). The following gradient system was applied: 0–2 min 20% B, 2–10 min 100% B, 10–12 min 100% B, and 12–15 min 20% B (balancing column until the initial conditions were restored). Chromatographic separation was conducted with a constant flow of the mobile phase (400 µL/min) at 30 °C.

#### 4.4.4. ICP-MS Method

A quadrupole ICP-MS 7800 (Agilent Technologies, Japan) equipped with an octupole collision cell was used for all 54 analysed trace elements. An internal standard (indium) was added to compensate for any effects from acid or instrument drift. The measurements were made with a platinum sampler and skimmer cones. The ICP-MS operational conditions are summarized in Appendix A.

#### 4.4.5. AAS Method

The measurements of Na, K and Ca were performed with an atomic absorption spectrometer: Solar GF Zeeman (Thermo Elemental) and iCE3500 (Thermo Scientific) equipped with single element hollow cathode lamps using an air/acetylene flame for the determination of Na and K and a nitrous oxide/acetylene flame for Ca. The wavelengths used for monitoring Na, K, and Ca were 766.5, 589.0 and 422.7 nm, respectively.

#### 4.4.6. NMR Spectroscopy

The spectroscopy method conditions were the same as those previously published [4]. The NMR spectra were recorded at 298 K on a Varian VNMRS-500 or Varian VNMRS-600 spectrometer equipped with a 5-mm Z-SPEC Nalorac IDG 500-5HT gradient probe or a 5-mm PFG AutoXID (1H/X15N-31P) probe, respectively.

## 5. Conclusions

Data on the formation of oxysterols and oxyphytosterols were available for over 20 years. However, based on the increasing sensitivity of mass spectrometers, an increasing number of compounds can be identified and quantified in samples. Using organic synthesis, various oxidized 3β,3′β-disteryl ethers were prepared, with structural confirmation by NMR and HRMS, and used to develop the chromatography method and optimise mass spectrometry for sensitive analysis. Indeed, most of the synthesized compounds were not detected in heated samples. However, the obtained results prove that oxidized 3β,3′β-disteryl ethers can indeed be formed through high-temperature treatment of sterol-rich samples, and the obtained results provide another portion of the information on reactions of sterol during high-temperature treatment. Additionally, to the best of our knowledge, this is the first report that describes the synthesis with NMR confirmation of oxidized 3β,3′β-disteryl ethers and proves that metal ions can catalyse the formation of 3β,3′β-disteryl ethers and their oxidized forms.

## Figures and Tables

**Figure 1 ijms-22-10421-f001:**
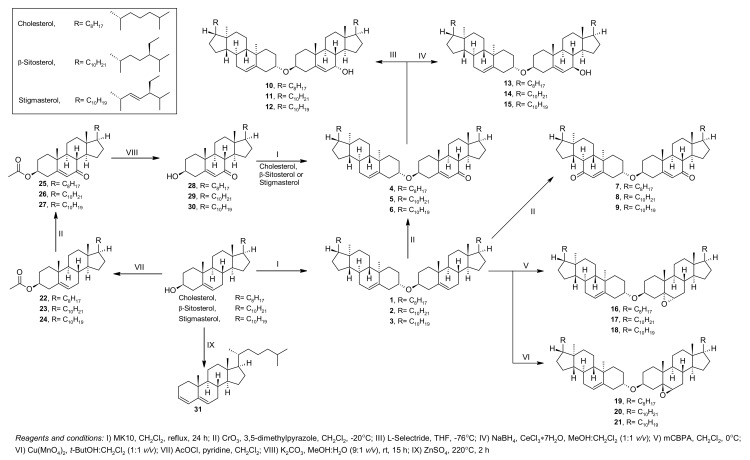
Synthesis of sterol derivatives.

**Figure 2 ijms-22-10421-f002:**
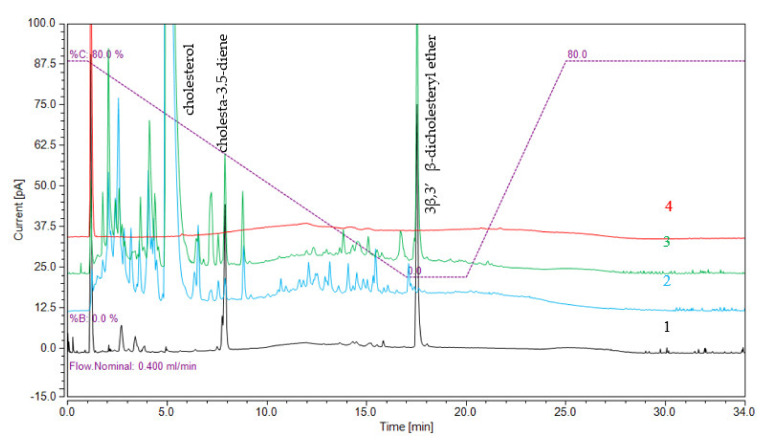
CAD chromatogram of analysed high-temperature samples with inorganic salts: (**1**) standard solution of cholesta-3,5-diene and 3β,3′β-dicholesteryl ether, (**2**) cholesterol mixed with calcium salt and heated for 0.5 h at 180 °C, (**3**) cholesterol mixed with copper salt and heated for 0.5 at 180 °C, and (**4**) blank solution (THF).

**Figure 3 ijms-22-10421-f003:**
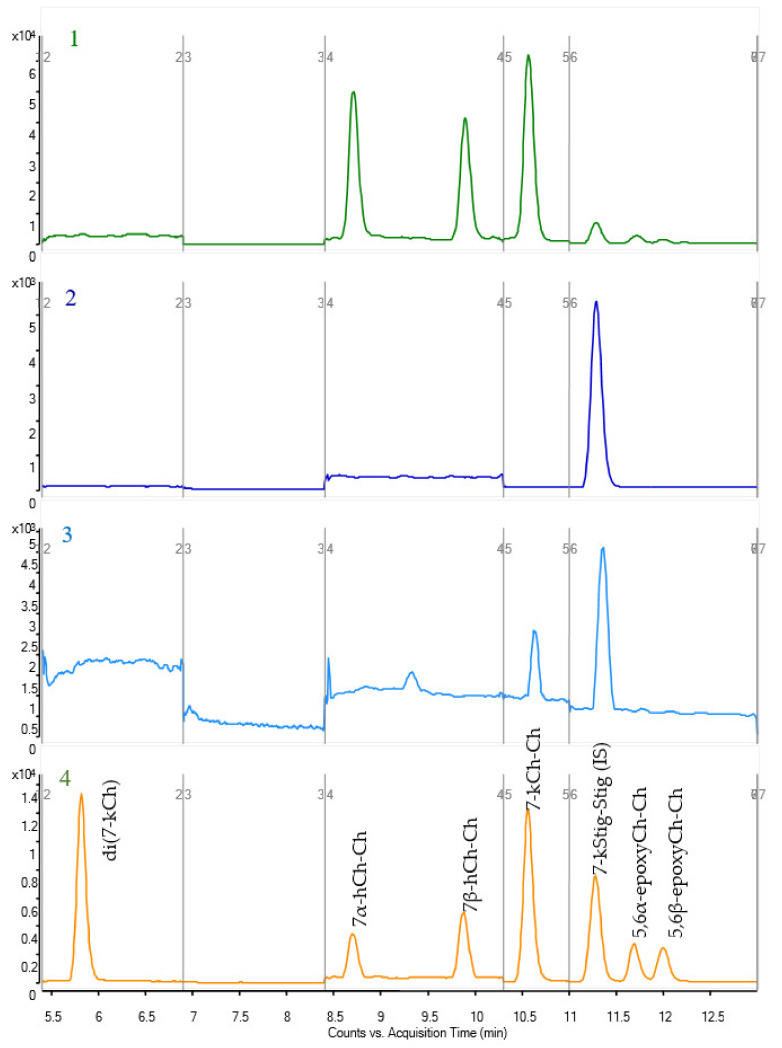
Total ion current (TIC) chromatogram of analysed oxidized 3β,3′β-dicholesteryl ethers: (**1**) sample after 3 h heating at 200 °C with magnesium, (**2**) blank with IS, (**3**) butter sample after 3 h heating at 200 °C, and (**4**) standard solution of all synthesized oxidized 3β,3′β-dicholesteryl ethers at a 100 ng/mL concentration.

**Table 1 ijms-22-10421-t001:** Measured content of 3β,3′β-dicholesteryl ether, cholesta-3,5-diene and oxidized 3β,3′β-dicholesteryl ethers in cholesterol samples with inorganic salts after heating at different temperatures and times.

Catalyst Addition(*n* = 3)	Time (h)	DCh	Cholestadiene	Cholesterol	7-KCh-Ch	Di-(7-KCh)	7α-HCh-Ch	7β-HCh-Ch	5,6α-EpoxyCh-Ch	5,6β-EpoxyCh-Ch
		(mg/g)		(%)	(µg/g)					
		180 °C								
CuSO_4_	0.5	17.29 ± 0.69a	7.61 ± 0.94a	96.28 ± 10.84a	0.84 ± 0.23a	<LOD	<LOD	<LOD	<LOD	<LOD
	1	47.00 ± 5.12b	43.03 ± 8.92b	75.52 ± 9.23a	0.54 ± 0.08a	<LOD	<LOD	<LOD	<LOD	<LOD
	2	80.51 ± 2.54c	101.38 ± 4.70c	37.57 ± 2.00b	0.62 ± 0.13a	<LOD	<LOD	<LOD	<LOD	<LOD
	3	103.15 ± 29.20d	127.53 ± 8.23d	28.67 ± 1.58c	0.94 ± 0.32a	<LOD	<LOD	<LOD	<LOD	<LOD
	4	102.09 ± 28.45d	139.59 ± 3.45d	21.28 ± 0.91d	1.52 ± 0.52a	<LOD	<LOD	<LOD	<LOD	<LOD
CaSO_4_	0.5	<LOD	<LOD	90.36 ± 0.39a	<LOD	<LOD	<LOD	<LOD	<LOD	<LOD
	1	<LOD	<LOD	80.87 ± 3.55b	<LOD	<LOD	<LOD	<LOD	<LOD	<LOD
	2	<LOD	<LOD	78.65 ± 2.51b	<LOD	<LOD	<LOD	<LOD	<LOD	<LOD
	3	<LOD	<LOD	80.26 ± 5.31b	<LOD	<LOD	<LOD	<LOD	<LOD	<LOD
	4	<LOD	<LOD	73.85 ± 2.55c	<LOD	<LOD	<LOD	<LOD	<LOD	<LOD
MgCl_2_	0.5	<LOD	<LOD	96.99 ± 10.51a	<LOD	<LOD	<LOD	<LOD	<LOD	<LOD
	1	<LOD	<LOD	85.06 ± 23.03a	<LOD	<LOD	<LOD	<LOD	<LOD	<LOD
	2	1.06 ± 0.82a	15.72 ± 5.49a	68.37 ± 13.50a	0.39 ± 0.14a	<LOD	0.11 ± 0.06a	0.08 ± 0.05a	<LOD	<LOD
	3	2.01 ± 1.10a	23.39 ± 7.16a	72.19 ± 12.70a	0.63 ± 0.17b	<LOD	0.28 ± 0.016a	0.15 ± 0.05a	<LOD	<LOD
	4	1.69 ± 1.53a	26.65 ± 8.62a	55.26 ± 26.73a	0.53 ± 0.16b	<LOD	0.23 ± 0.15a	0.10 ± 0.05a	<LOD	<LOD
ZnSO_4_	0.5	7.93 ± 3.09a	<LOD	60.19 ± 11.28a	0.24 ± 0.19a	<LOD	<LOD	<LOD	<LOD	<LOD
	1	25.14 ± 5.34b	5.95 ± 1.87a	41.41 ± 5.46b	0.60 ± 0.25a	<LOD	<LOD	<LOD	<LOD	<LOD
	2	33.02 ± 6.69b	14.55 ± 5.98b	45.23 ± 5.19b	0.48 ± 0.31a	<LOD	<LOD	<LOD	<LOD	<LOD
	3	45.80 ± 14.84b	25.28 ± 11.65c	47.13 ± 5.66b	0.58 ± 0.12a	<LOD	<LOD	<LOD	<LOD	<LOD
	4	52.26 ± 9.07b	32.10 ± 7.82c	38.08 ± 6.13b	0.43 ± 0.18a	<LOD	<LOD	<LOD	<LOD	<LOD
AlCl_3_	0.5	12.91 ± 2.95a	125.79 ± 9.64a	48.54 ± 14.71a	0.78 ± 0.13a	<LOD	<LOD	<LOD	<LOD	<LOD
	1	10.98 ± 1.83a	159.67 ± 13.99a	19.30 ± 4.15b	0.58 ± 0.06a	<LOD	<LOD	<LOD	<LOD	<LOD
	2	10.78 ± 2.32a	162.07 ± 28.63a	<LOD	0.49 ± 0.03a	<LOD	<LOD	<LOD	<LOD	<LOD
	3	9.83 ± 2.26a	182.29 ± 53.17a	<LOD	0.33 ± 0.06b	<LOD	<LOD	<LOD	<LOD	<LOD
	4	8.65 ± 3.30a	183.78 ± 47.15a	<LOD	0.22 ± 0.03c	<LOD	<LOD	<LOD	<LOD	<LOD
		**200 °C**								
CuSO_4_	0.5	59.98 ± 19.35a	62.13 ± 15.5a	51.07 ± 10.77a	2.08 ± 0.32a	<LOD	<LOD	<LOD	<LOD	<LOD
	1	71.87 ± 3.86a	124.72 ± 41.91b	19.51 ± 13.20b	2.38 ± 0.15a	<LOD	<LOD	<LOD	<LOD	<LOD
	2	80.63 ± 10.03a	187.91 ± 7.76c	15.44 ± 12.46b	2.32 ± 0.25a	<LOD	<LOD	<LOD	<LOD	<LOD
	3	69.36 ± 6.38a	212.16 ± 14.51c	<LOD	2.37 ± 0.02a	<LOD	<LOD	<LOD	<LOD	<LOD
	4	39.28 ± 10.95b	129.82 ± 7.69d	<LOD	2.89 ± 0.30b	<LOD	<LOD	<LOD	<LOD	<LOD
CaSO_4_	0.5	<LOD	10.89 ± 3.55a	77.75 ± 0.76a	<LOD	<LOD	<LOD	<LOD	<LOD	<LOD
	1	<LOD	19.95 ± 3.04b	63.19 ± 6.72b	0.15 ± 0.03a	<LOD	<LOD	<LOD	<LOD	<LOD
	2	<LOD	5.12 ± 1.01c	61.84 ± 14.46b	0.18 ± 0.05a	<LOD	<LOD	<LOD	<LOD	<LOD
	3	<LOD	<LOD	50.21 ± 6.94b	0.02 ± 0.00b	<LOD	<LOD	<LOD	<LOD	<LOD
	4	<LOD	<LOD	52.11 ± 15.68b	<LOD	<LOD	<LOD	<LOD	<LOD	<LOD
MgCl_2_	0.5	1.04 ± 0.91a	3.76 ± 1.58a	83.53 ± 12.80a	<LOD	<LOD	<LOD	<LOD	<LOD	<LOD
	1	11.51 ± 2.13b	35.10 ± 3.89b	55.62 ± 9.77b	0.73 ± 0.18a	<LOD	0.13 ± 0.11a	0.06 ± 0.04a	<LOD	<LOD
	2	19.79 ± 2.83c	34.90 ± 9.35b	35.90 ± 5.55c	4.62 ± 1.72b	0.09 ± 0.05a	0.75 ± 0.31b	0.34 ± 0.17b	0.09 ± 0.04a	<LOD
	3	19.32 ± 3.12c	54.80 ± 4.62c	24.24 ± 2.54d	5.65 ± 1.78b	0.17 ± 0.05b	1.12 ± 0.39b	0.47 ± 0.33b	0.17 ± 0.07a	<LOQ
	4	28.77 ± 4.09d	91.47 ± 5.04d	24.42 ± 5.29d	5.29 ± 2.05b	0.16 ± 0.11b	1.18 ± 0.47b	0.56 ± 0.16b	0.17 ± 0.04a	0.02 ± 0.01
ZnSO_4_	0.5	8.97 ± 2.24a	21.47 ± 4.42a	65.91 ± 11.85a	0.70 ± 0.13a	<LOD	<LOD	<LOD	<LOD	<LOD
	1	11.02 ± 1.08a	36.76 ± 9.72a	61.85 ± 16.70a	0.92 ± 0.65a	<LOD	<LOD	<LOD	<LOD	<LOD
	2	18.32 ± 3.58b	94.69 ± 9.24b	47.68 ± 8.93a	1.47 ± 0.21a	<LOD	<LOD	<LOD	<LOD	<LOD
	3	18.86 ± 2.33b	122.82 ± 13.98c	42.19 ± 8.29a	1.61 ± 0.025a	<LOD	<LOD	<LOD	<LOD	<LOD
	4	20.86 ± 6.08b	178.80 ± 30.38d	39.24 ± 6.14a	0.71 ± 0.32b	<LOD	<LOD	<LOD	<LOD	<LOD
AlCl_3_	0.5	17.49 ± 1.23a	157.45 ± 38.68a	<LOD	1.87 ± 0.58a	<LOD	0.67 ± 0.05a	0.47 ± 0.01a	0.14 ± 0.04a	<LOD
	1	16.57 ± 2.25a	129.16 ± 36.05a	<LOD	1.24 ± 0.09a	<LOD	0.32 ± 0.07b	0.27 ± 0.06b	0.06 ± 0.02b	<LOD
	2	12.86 ± 2.42a	88.53 ± 21.36a	<LOD	0.34 ± 0.05b	<LOD	<LOD	<LOD	<LOD	<LOD
	3	12.24 ± 2.61a	137.43 ± 58.57a	<LOD	0.23 ± 0.02c	<LOD	<LOD	<LOD	<LOD	<LOD
	4	11.75 ± 3.31a	131.82 ± 44.11a	<LOD	0.18 ± 0.02d	<LOD	<LOD	<LOD	<LOD	<LOD
		**220 °C**								
CuSO_4_	0.5	51.59 ± 13.79a	141.66 ± 31.36a	25.41 ± 6.50	2.81 ± 0.21a	<LOD	<LOD	<LOD	<LOD	<LOD
	1	39.32 ± 12.67a	180.62 ± 66.02a	<LOD	3.07 ± 0.42a	<LOD	<LOD	<LOD	<LOD	<LOD
	2	34.55 ± 16.65a	150.35 ± 82.61a	<LOD	2.53 ± 0.54a	<LOD	<LOD	<LOD	<LOD	<LOD
	3	13.49 ± 2.81b	146.79 ± 22.55a	<LOD	1.14 ± 0.69b	<LOD	<LOD	<LOD	<LOD	<LOD
	4	<LOD	115.87 ± 52.07a	<LOD	0.34 ± 0.10c	<LOD	<LOD	<LOD	<LOD	<LOD
CaSO_4_	0.5	<LOD	<LOD	75.08 ± 5.72a	<LOD	<LOD	<LOD	<LOD	<LOD	<LOD
	1	<LOD	11.78 ± 5.30a	57.63 ± 2.68b	0.13 ± 0.04a	<LOD	<LOD	<LOD	<LOD	<LOD
	2	<LOD	27.31 ± 6.35b	63.64 ± 9.53b	0.04 ± 0.02b	<LOD	<LOD	<LOD	<LOD	<LOD
	3	<LOD	57.77 ± 10.61c	47.59 ± 11.41c	0.06 ± 0.02b	<LOD	<LOD	<LOD	<LOD	<LOD
	4	<LOD	68.92 ± 14.18c	52.92 ± 3.01c	0.04 ± 0.02b	<LOD	<LOD	<LOD	<LOD	<LOD
MgCl_2_	0.5	13.04 ± 4.21a	15.96 ± 11.91a	41.77 ± 4.97a	3.21 ± 0.97a	0.08 ± 0.01a	0.85 ± 0.43a	0.37 ± 0.19a	0.08 ± 0.01a	<LOD
	1	12.73 ± 3.86a	83.48 ± 6.72b	14.70 ± 9.95b	3.98 ± 0.62a	0.12 ± 0.04a	1.15 ± 0.71a	0.48 ± 0.19a	0.08 ± 0.02a	<LOD
	2	8.71 ± 6.49a	187.61 ± 30.91c	<LOD	0.80 ± 0.29b	<LOD	0.27 ± 0.24b	0.17 ± 0.16a	<LOD	<LOD
	3	<LOD	33.25 ± 19.87d	<LOD	<LOD	<LOD	<LOD	<LOD	<LOD	<LOD
	4	<LOD	26.00 ± 0.66d	<LOD	<LOD	<LOD	<LOD	<LOD	<LOD	<LOD
ZnSO_4_	0.5	16.58 ± 4.80a	57.24 ± 5.49a	79.64 ± 8.13a	0.43 ± 0.08a	<LOD	<LOD	<LOD	<LOD	<LOD
	1	19.30 ± 3.60a	136.51 ± 27.11b	58.31 ± 8.96b	0.99 ± 0.026b	<LOD	<LOD	<LOD	<LOD	<LOD
	2	23.19 ± 1.82a	205.20 ± 35.28c	39.10 ± 3.01c	0.94 ± 0.37b	<LOD	<LOD	<LOD	<LOD	<LOD
	3	22.18 ± 3.12a	260.49 ± 10.75d	37.45 ± 4.50c	0.68 ± 0.011cc	<LOD	<LOD	<LOD	<LOD	<LOD
	4	24.91 ± 5.54a	320.79 ± 6.19e	21.71 ± 9.44d	<LOD	<LOD	<LOD	<LOD	<LOD	<LOD
AlCl_3_	0.5	13.52 ± 0.58a	140.48 ± 17.22a	<LOD	0.40 ± 0.18a	<LOD	<LOD	<LOD	<LOD	<LOD
	1	8.11 ± 0.38b	103.80 ± 24.13a	<LOD	0.21 ± 0.13a	<LOD	<LOD	<LOD	<LOD	<LOD
	2	3.29 ± 0.55c	41.30 ± 2.30b	<LOD	0.04 ± 0.03b	<LOD	<LOD	<LOD	<LOD	<LOD
	3	0.86 ± 0.17d	17.82 ± 1.94c	<LOD	<LOD	<LOD	<LOD	<LOD	<LOD	<LOD
	4	<LOD	<LOD	<LOD	<LOD	<LOD	<LOD	<LOD	<LOD	<LOD

Values within a column with different letters are significantly different (*p* < 0.05).

**Table 2 ijms-22-10421-t002:** Measured content of oxidized 3β,3′β-disteryl ethers in chosen biological samples.

		Measured Content [ng/g]				
	Time (h)	0.5	1	2	3	4
Sample	Compound					
Temperature		180 °C				
**Butter**	7-kCh-Ch	<LOD	<LOD	<LOD	18.17 ± 0.99a	40.53 ± 13.73b
**Cod-liver oil**	7-kCh-Ch	<LOD	<LOD	<LOQ	<LOQ	39.01 ± 12.05
**Rapeseed oil**	7-kSito-Sito	2399.33 ± 560.95a	105.62 ± 43.04b	23.51 ± 15.82c	<LOD	<LOD
	7-kStigm-Stigm	<LOD	<LOD	<LOD	<LOD	<LOD
**Corn oil**	7-kSito-Sito	<LOD	<LOD	<LOD	<LOD	<LOD
	7-kStigm-Stigm	<LOD	<LOD	<LOD	<LOD	<LOD
		**200 °C**				
**Butter**	7-kCh-Ch	<LOD	<LOD	<LOD	<LOQ	23.62 ± 6.99
**Cod-liver oil**	7-kCh-Ch	29.49 ± 6.30	<LOD	<LOD	<LOD	<LOD
**Rapeseed oil**	7-kSito-Sito	<LOD	<LOD	<LOD	<LOD	<LOD
	7-kStigm-Stigm	<LOD	<LOD	<LOD	<LOD	<LOD
**Corn oil**	7-kSito-Sito	<LOD	<LOD	<LOD	<LOD	<LOD
	7-kStigm-Stigm	<LOD	<LOD	<LOD	<LOD	<LOD
		**220 °C**				
**Butter**	7-kCh-Ch	<LOD	<LOD	<LOD	<LOD	<LOD
**Cod-liver oil**	7-kCh-Chr	<LOD	<LOD	<LOD	<LOD	<LOD
**Rapeseed oil**	7-kSito-Sito	<LOD	<LOD	<LOD	<LOD	<LOD
	7-kStigm-Stigm	<LOD	<LOD	<LOD	<LOD	<LOD
**Corn oil**	7-kSito-Sito	<LOD	<LOD	<LOD	<LOD	<LOD
	7-kStigm-Stigm	<LOD	<LOD	<LOD	<LOD	<LOD

Values within a row with different letters are significantly different (*p* < 0.05).

## Data Availability

Not applicable.

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
