# Peer review of "Synthesis of Oxidized 3β,3′β-Disteryl Ethers and Search after High-Temperature Treatment of Sterol-Rich Samples"

_ijms, 2021, doi:10.3390/ijms221910421_

Round 1
Reviewer 1 Report
In this manuscript the authors prepared and structurally characterized different oxidized 3β,3’β-disteryl ethers. They performed quantitative analysis of the oxidized 3β,3’β-disteryl ethers by using liquid extraction, solid-phase extraction and liquid chromatography coupled with mass spectrometry. They also investigated the influence of metal compounds on the mechanism of ether formation at high temperatures.
In my opinion, the experiments are performed well and authors' conclusions are sound. Overall, I recommend the publication of this manuscript after revision considering the minor point addressed below:
Figure 3 in the current form of the manuscript should be placed as Figure 1 in the beginning of the Results section where authors describe the results of the synthesis of the sterol derivatives. In addition, structures of some compounds which are mentioned in the section 2.1. (e.g. 32 and 33) are not visible in the synthesis route. Authors should correct this.
Author Response
Remark 1: Figure 3 in the current form of the manuscript should be placed as Figure 1 at the beginning of the Results section where the authors describe the results of the synthesis of the sterol derivatives.
Author response: The order of the Figures has been corrected.
Remark 2. In addition, structures of some compounds which are mentioned in the section 2.1. (e.g. 32 and 33) are not visible in the synthesis route.
Author response: there were errors in compounds numbers. Section 2.1 was corrected.
Additionally, one of the authors requested a change in the affiliation. The proper changes were made.
Reviewer 2 Report
In this manuscript, the authors attempt to clarify the different reactions of sterols at high temperature. Especially the apparition of oxidated compounds and dimers is studied as a continuation of previous publications.
The synthesis of several possible molecules, monomers, and dimers in different stages of oxidation was carried out for its use as standards in the analysis of the products resulting from the thermal treatment of sterols. Those synthetic procedures are correct and the prepared compounds are well characterized. The NMR spectra show a good degree of purity on all samples.
The analytical procedures used for the quantification of the compounds formed in the thermal experiments are well described.
The conclusions agree with the data presented.
Thus, this manuscript contains interesting information, although somewhat limited to a very specific area, and can be published.
There are some minor mistakes in the supplementary Materials that should be fixed:
Table S9. The caption says “…2, 5 and 35 steroids”, and it should be “…2, 5 and 29 steroids”
Table S10 same mistake. Says “…3, 6 and 36 steroids” instead of “…3, 6 and 30 steroids”
Figs. S36, S42, and S45 are missing, at least in the downloaded copy. Should be checked.
Author Response
Remark 1: Table S9. The caption says “…2, 5 and 35 steroids”, and it should be “…2, 5 and 29 steroids”
Author response: Table S9 has been corrected.
Remark 2: Table S10 same mistake. Says “…3, 6 and 36 steroids” instead of “…3, 6 and 30 steroids”
Author response: Table S10 has been corrected.
Remark 3: Figs. S36, S42, and S45 are missing, at least in the downloaded copy. Should be checked.
Author response: Missing Figures has been added to the supplementary data.
Additionally, one of the authors requested a change in the affiliation. The proper changes were made.